REGISTERED REPORT PROTOCOL

# Effects of internet-based telemonitoring platforms on the quality of life of oncologic patients: A systematic literature review protocol

**Felipe Martínez**[1,2,3]*, **Catalina Tobar**[1,2,4], **Carla Taramasco**[1,5]

**1** Centro para la Prevención y Control del Cáncer (CECAN), Santiago, Chile, **2** Concentra Educación e Investigación Biomédica, Viña del Mar, Chile, **3** Facultad de Medicina, Escuela de Medicina, Universidad Andrés Bello, Viña del Mar, Chile, **4** Servicio de Medicina, Hospital Gustavo Fricke, Viña del Mar, Chile, **5** Facultad de Ingeniería, Universidad Andrés Bello, Viña del Mar, Chile

* felipe.martinez@concentrainvestigacion.cl

## Abstract

### Introduction

Telemonitoring involves the transmission of clinical information through digital means, including internet-connected devices such as smartphones, health tracking apps and video conferencing platforms. This strategy could provide a viable alternative to facilitate follow-up in several conditions, including cancer.

### Objectives

To synthesise the available evidence on the effectiveness of internet-based telemonitoring platforms amongst oncological patients. Relevant endpoints include overall quality of life, the ability to detect postoperative complications, severe toxicity reactions attributable to che-motherapy, reducing the frequency of hospitalisations, emergency department visits and mortality.

### Methods

A systematic review of published and unpublished randomised and controlled studies will be carried out. Iterative searches in PubMED/MEDLINE, EMBASE, Epistemonikos, LILACS, and Cochrane CENTRAL repositories from January 2000 to January 2023 will be conducted. Grey literature repositories, such as Clinicaltrials, BioRxiv and MedRxiv will be searched as well. The Cochrane risk of bias tool will be used to assess the quality of the eligible studies. If possible, a meta-analysis based on the random-effects model will be conducted to evaluate changes in any of the aforementioned outcomes. Heterogeneity will be assessed with Cochrane's Q and I2 statistics. Its exploration will be carried out using subgroup and sensitivity analyses. Relevant subgroups include the proportion of elderly patients in each study, characteristics of each platform, study type, type of funding and

**Funding:** This work has been funded by the Agencia Nacional de Investigación y Desarrollo (ANID) as part of the Fondo de Financiamiento de Centros de Investigación en Áreas Prioritarias research initiative. ANID FONDAP 152220002 (CECAN)

moment of conduction (i.e. before or after the COVID-19 pandemic). Publication bias will be assessed using funnel plots and Egger's test.

## Registration

This systematic review protocol is registered in PROSPERO. Its registration number is CRD42023412705.

## Introduction

Cancer is one of the leading causes of death worldwide, accounting for nearly 10 million deaths in 2020 according to data from the World Health Organisation [1] This increase can be attributed to the ageing and expansion of the population, accompanied by shifts in the prevalence and distribution of major cancer risk factors. Notably, many of these risk factors are intertwined with socioeconomic development. The most common forms of cancer are breast, lung, colon, rectum and prostate cancers. The global burden of cancer incidence and mortality is escalating at a rapid pace. However, several advancements in treatments have improved the overall prognosis of many forms of cancer. On the other hand,, treatments such as chemotherapy or radiotherapy can cause a vast array of side effects such as nausea, pain, fatigue, and diarrhoea and can even become life-threatening, such as with cases of neutropenic fever and sepsis [2]. In most cases, treatment for cancer is currently provided in an ambulatory setting, which requires careful consideration and planning of a follow-up strategy [3]. However, there is remarkable heterogeneity in clinical practice regarding follow-up practices which vary from centre to centre.

Traditional follow-up methods for cancer patients face several limitations. They typically adhere to fixed schedules, often incongruent with patients' needs. These methods are resource-intensive and may strain healthcare systems. Geographical barriers can impede access to follow-up care, especially for remote or rural residents, and financial burdens, stemming from travel costs and time off work, deter some patients from attending follow-up appointments. These methods lack continuous patient engagement between appointments and might not be tailored to individual risks, potentially leading to over- or under-treatment. Overcoming these issues requires a more personalised, patient-centric approach with flexible scheduling and improved accessibility.

Remote monitoring, or telemonitoring, is becoming an increasingly popular alternative for maintaining surveillance of several medical conditions, including cancer [4–7]. Briefly, telemonitoring involves the transmission of relevant clinical information through digital means, which include internet-connected medical devices such as smartphones, health tracking apps and video conferencing platforms. These tools not only allow for the surveillance of various clinical conditions, but also represent an alternative for maintaining communication between patients and doctors [8, 9]. The main idea is to provide accurate and timely medical care, which can improve the patient's quality of life and reduce treatment costs by eliminating the need for regular in-person visits. In this sense, telemonitoring platforms represent particularly interesting tools for people who have difficulty achieving in-person medical check-ups, as well as those who have to travel long distances to be able to carry out such visits. Additionally, it is likely that the need for fewer visits to healthcare centres makes for a less disruptive daily life for the patient, which can facilitate their social integration and improve their quality of life [6, 10].

Previous experiences among adult cancer patients have shown that various telemonitoring strategies are feasible to implement and can effectively contribute to patient care. Initial studies have shown improvements in the detection of symptoms such as pain [11, 12], quality of life [13], and the detection of complications following surgical interventions [14, 15]. While these initial experiences are promising regarding the use of these strategies, their exploratory and pilot nature generally restricts their weight when it comes to defining definitive conclusions. Considering the significant epidemiological importance of cancer for the population and the potential contribution that this type of intervention could have, it is necessary to conduct a synthesis of the available evidence on this topic that allows for obtaining relevant information for the execution of clinical studies.

## Objectives

The primary objective of this study is to synthesise the available evidence regarding the effectiveness of internet-based telemonitoring platforms compared to standard in person follow-up strategies of adult cancer patients. In particular, we aim to determine whether these strategies can improve any of the following clinical outcomes:

- Improving the quality of life of adult cancer patients

- Improving the ability to detect postoperative complications in adult cancer patients

- Improving the ability to detect severe toxicity reactions attributable to chemotherapy

- Reducing the frequency of hospitalizations among adult cancer patients

- Reducing the frequency of emergency department visits among adult cancer patients

- Reducing mortality among adult cancer patients

## Methods

To fulfil the aforementioned objectives, a systematic review of the literature of published and unpublished studies regarding the effects of telemonitoring platforms among adult cancer patients will be conducted. This systematic review protocol has been designed following the *Preferred reporting items for systematic review and meta-analysis protocols* recommendations [16] and will follow the standard methodological norms established by the Cochrane Collaboration [17]. This protocol has been registered in the *International Prospective Register of Systematic Reviews (PROSPERO)* at the University of York. Its registration number is CRD42023412705. The funder had no role in the conception nor design of the protocol.

### Search strategy

An iterative bibliographic search will be carried out in the PubMED/MEDLINE, EMBASE, Epistemonikos, LILACS, and Cochrane CENTRAL repositories from January 2000 to January 2023. This temporal restriction was considered since Internet-based telemonitoring technologies represent relatively recent interventions. No language restrictions will be applied within this search strategy. The results of the search strategy will be synthesised using a PRISMA flowchart [18], as shown in Fig 1.

**Elegibility criteria.** Primary studies evaluating telemonitoring platforms among adult (>18 years old) patients with any type of cancer receiving outpatient treatment and containing data for any of the outcomes of interest in this systematic review will be considered for

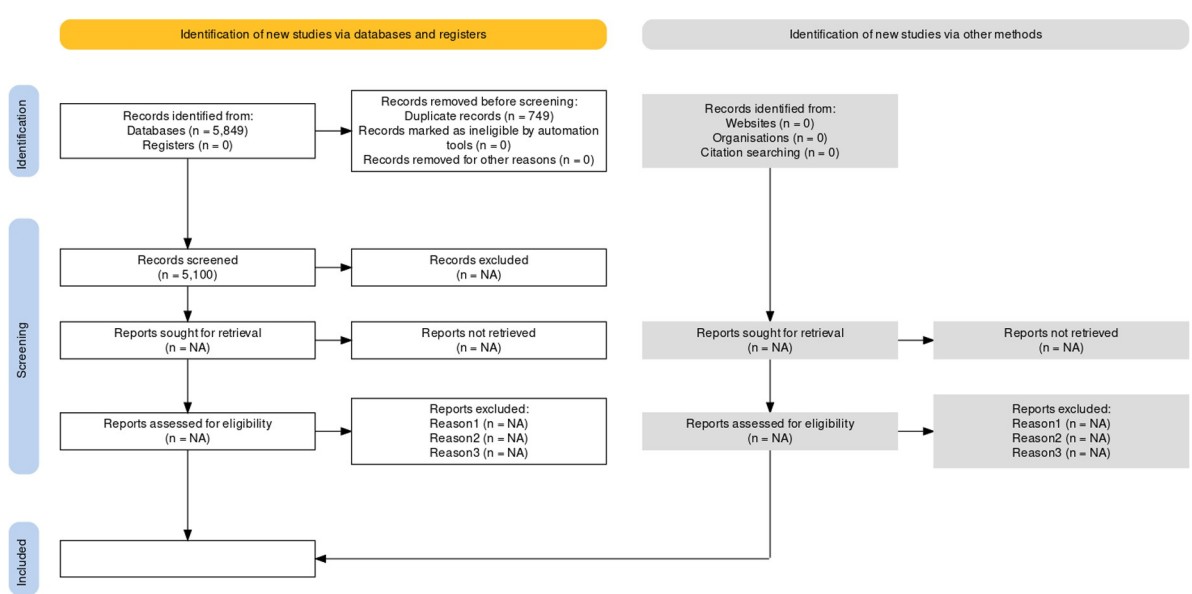

**Fig 1. PRISMA flowchart depicting the search strategy for the systematic review.**

inclusion. Any intervention involving the transmission of clinical information via the internet using digital media, including internet-connected medical devices, smartphones, health tracking applications, and videoconferencing platforms, with the aim of communicating this information to clinical teams will be considered a telemonitoring strategy. Studies also had to use traditional in-person follow-up methods as a comparator in order to be eligible for this systematic review. Studies that do not report data for any of the study outcomes of the review will be excluded, as will those assessing telemonitoring interventions amongst children or mixed populations that include paediatric patients and those comparing different telemonitoring strategies or media (i.e. telephone-based monitoring vs. internet-based monitoring).

Given the interventional nature of the clinical question posed, only randomised controlled trials and controlled clinical trials will be considered to answer the research questions. If insufficient studies of the aforementioned designs are found in initial searches, the scope will be widened to include before and after studies as well. A bibliographic search for unpublished articles will also be conducted by reviewing repositories of clinical trial protocols (such as ClinicalTrials.gov), and preprint article repositories for health sciences, including medRxiv and bioRxiv via EMBASE.

The search strategy will be designed and carried out by a research librarian with more than 5 years of experience in conducting literature searches for systematic reviews in healthcare. A stepwise approach will be used for study inclusion in the review. This approach involves selecting studies based on their relevance established from the analysis of titles, and then proceeding to a second filter for evaluation of essential methodological aspects and population adequacy based on abstract analysis. Potentially relevant texts will be selected from this procedure for full-text evaluation. If there is no agreement between the authors regarding the inclusion of a candidate study, a third author (CT) will be asked to act as an arbitrator to determine eventual inclusion or exclusion of the study. The reason for exclusion of each work will be recorded on an annex form. A review of the reference lists of each included study will also be carried out to complement the search strategy.

The bibliographic search will be carried out iteratively by implementing the application of search terms classified into the domains Patient, Intervention, Comparator, and Outcome.

The combination of these search terms will be carried out using boolean operators "AND" to combine domains and "OR" for terms belonging to the same domain. Search results will be handled using Rayyan [19], an user-friendly online platform that facilitates the process of duplicate detection, removal and assists in the conduction of blind assessments of study eligibility amongst independent reviewers. The complete search strategy including specific terms and their combination is provided in the **S1 Appendix.**

## Study outcomes

In this systematic review, two types of outcomes, primary and secondary, will be considered. Primary outcomes represent events of great importance in the care of cancer patients that are relevant across different cancer types and treatments. Secondary outcomes, on the other hand, will correspond to all those that apply to narrower populations within cancer patients [for example, only those receiving surgical treatment or systemic chemotherapy] or whose intensity is of secondary relevance in the prognosis of the disease.

Main outcomes for this systematic review include:

- *Quality of life (QoL)*: Quality of life pertains to a person's overall well-being and satisfaction. In this systematic review, only validated tools reporting quality of life estimates will be considered, such as the EQ-5D questionnaire.

- *Rate of Hospitalisations*: Admission to a hospital is a major event for patients receiving treatment for cancer. Telemonitoring platforms might help detect complications in early stages, thus allowing for timely interventions to be made that might avoid hospitalisation. The proportion of patients requiring an unplanned hospitalisation will be considered as the outcome measure for this endpoint.

- *Mortality*: This outcome will be addressed as the proportion of patients that die during follow-up.

On the other hand, secondary outcomes for this systematic review comprise:

- *Severe toxicity reactions attributable to chemotherapy*: Chemotherapy-associated toxicity can greatly influence the course of treatment for a patient with cancer. Any adverse event attributable to chemotherapy that limits self-care or mandates hospitalisation will be considered a severe adverse event. The outcome measure for this endpoint will be the proportion of patients that develop a severe toxicity event attributable to chemotherapy.

- *Postoperative complications*: Postoperative complications following cancer surgery can vary depending on the type and extent of the surgery, the specific type of cancer being treated, the patient's overall health, and other individual factors. Given this heterogeneity, this outcome will be addressed as the proportion of participants that develop any postoperative complication. Postoperative complications that result in hospital admission will be labelled as severe.

- *Emergency department visits*: This outcome will be expressed as the proportion of participants in each study arm that required an unscheduled emergency department visit during follow-up.

## Quality assessment of included studies

After the initial screening and consideration of the aforementioned inclusion criteria, all studies will be independently assessed for their methodological quality by two reviewers based on

standard criteria (CTB and FM). In case of disagreement regarding the methodological quality of an individual study, a third author will act as an arbitrator to resolve the evaluation (CT). Randomised or controlled clinical trials will be evaluated using the criteria proposed by the Cochrane Collaboration to assess the quality of interventional studies [17]. These criteria include concealment allocation sequences, the level of masking used in each trial, the number of patients lost during follow-up, and the analysis strategy selected (intention-to-treat principle, per-protocol or other). An open section will also be provided for other sources of systematic error that may be relevant to the interpretation of each study's findings.

## Certainty of the evidence

Outside of the aforementioned critical analysis, the GRADE methodology will be implemented to evaluate the certainty of the available evidence regarding the usefulness of internet-based telemonitoring platforms for clinical surveillance of cancer patients [20, 21]. This procedure will also be carried out by two independent authors (CTB, FM), leaving a third author as an arbiter in case of not reaching a consensus. Briefly, the GRADE methodology assigns a score to establish the certainty of the evidence. Among the factors that decrease this certainty are the perceived risk of bias, inconsistencies in the magnitude of the effect associated with heterogeneity, indirect estimation of the clinical effect, imprecision of the estimators, and the possibility of publication bias. Each of these elements deducts between one and two points based on the reviewers' perception of uncertainty. There are also factors that increase the certainty of the evidence, including signs of a dose-response effect, a large effect size, and the exclusion of the possibility of residual confounding. These results can be synthesised into a Summary of Findings table integrated with the detected effects.

## Data extraction

Two independent authors (CTB, FM) will extract information from each included study using a standard form. In case of any disagreements regarding the extracted data, the researchers will first resolve them through discussion. If a conclusion cannot be reached, a third reviewer will act as an arbitrator to determine the data to be included. In cases where the required information cannot be obtained from the reported results of an included study, an attempt will be made to contact the authors for additional information, if available.

The information to be collected within this systematic review will be grouped into three essential domains. The first domain corresponds to the study characteristics, which will contain clinical and demographic data to describe the general environment in which the study intervention takes place, thus allowing for a better characterisation of the intervention's applicability. All this information will be recorded in addition to aspects that allow for the assessment of study quality. Considering the non-pharmacological nature of the intervention, information will also be gathered regarding the mode of implementation of telemonitoring tools, aiming to inform about available alternatives and potentially identify characteristics associated with their success. Finally, information from the third domain related to the study outcomes will be included. Specifically, data to be obtained from each study will include:

- **Study Characteristics**

  ○ Study type (randomised clinical trial, controlled clinical trial, or before-and-after study).

  ○ Study date (year of publication).

  ○ Country where the study was conducted.

  ○ Number of participants (total and per group).

○ Age range of participants (years).

○ Source of funding for the study (public, industry, independent).

○ Type of cancer (specific and then categorised into groups: haematological, solid, mixed).

○ Most prevalent cancer stage in the study based on TNM classification [22].

○ Type of treatment (chemotherapy, surgery, radiotherapy, mixed).

○ Proportion of elderly patients (n, %).

○ Proportion of patients with chronic kidney disease (n, %).

○ Proportion of patients with chronic liver damage (n, %).

○ Proportion of patients with cognitive impairment (n, %).

- **Intervention Characteristics:**

  ○ Type of platform implemented (smartphone application, video calls, other).

  ○ Duration of intervention usage (months).

  ○ Delivery of educational content in the intervention (Yes/No).

  ○ Provision of self-management strategies within the intervention (Yes/No).

  ○ Remote monitoring of vital signs by clinicians (Yes/No).

  ○ Remote monitoring of symptoms by clinicians (Yes/No).

  ○ Monitoring of symptoms by patients (Yes/No).

  ○ Provision of a platform to facilitate between-patient communication (Yes/No).

  ○ Provision of a platform to facilitate clinician-patient communication (Yes/No).

- **Study Endpoints:**

  ○ Type and results in quality of life scales within the study (mean and standard deviation, SD).

  ○ Incidence of postoperative complications between groups (n, %), in studies where the intervention was implemented amongst surgical patients.

  ○ Incidence of severe adverse reactions attributable to chemotherapy between groups in cases, where applicable (n, %).

  ○ Incidence of hospitalisations between groups (n, %).

  ○ Number of deaths that occurred between groups (n, %).

  ○ Number of reported emergency department visits between groups (n, %).

## Analysis plan

If appropriate, data will be synthesised in a meta-analysis. If data are not amenable to quantitative synthesis, a descriptive approach will be undertaken to summarise findings. A random-effects model will be preferred to conduct summary estimates considering the non-pharmacological nature of the intervention, which is expected to lead to greater heterogeneity between

individual studies. For binary outcomes, an estimation of the relative risk or odds ratio will be calculated, accompanied by its corresponding 95% confidence intervals. For continuous outcomes, such as the difference in quality of life scale scores, the difference in means will be used as a summary statistic associated with a 95% confidence interval. If different scales are used to evaluate outcomes (such as quality of life), a weighted mean difference will be implemented instead.

The heterogeneity of the results will be assessed using Cochrane's Q and I2 statistics. The I2 statistic provides a measurement of the variation between studies that cannot be attributed to chance and is expressed as a percentage. It is often categorised as follows: <25% low, 25 to 50% moderate, and >50% high[17, 23, 24]. The fixed-effect model will only be considered to statistically synthesise the results if they are homogeneous (I2<25%). Heterogeneity will also be explored through pre-specified subgroup analysis. Subgroup analysis has been considered based on the proportion of elderly patients in each study, the type and characteristics of each intervention implemented, the type of study (randomised vs. non-randomised), the moment in which the study was conducted (prior of after the COVID-19 pandemic), and the type of funding received by the researchers (private vs. public funding). Additionally, studies will be grouped based on their perceived methodological quality for conducting subgroup analyses.

Publication bias will be evaluated using a funnel plot and Egger's test. All analyses will be performed in Review Manager (RevMan) [computer program], version 5.2, Copenhagen: The Nordic Cochrane Centre, The Cochrane Collaboration, 2012. Review Manager (RevMan) is a software recommended by the Cochrane Collaboration for conducting systematic reviews and meta-analyses in evidence-based medicine. It aids study selection, data extraction, quality assessment, meta-analysis, and report generation, enhancing the efficiency and accuracy of evidence synthesis for informed healthcare decisions.

## Summary

This systematic review protocol will provide a summary of current evidence regarding the use of internet-based telemonitoring platforms amongst adult patients with cancer, which is a key step to further knowledge on this subject. If possible, the impact of this strategy in several relevant outcomes such as quality of life, overall survival and the development of several key complications during cancer treatment will be assessed in a meta-analysis that will assist healthcare providers considering telemonitoring as an option for patient surveillance.

## Supporting information

**S1 Appendix. Search strategy.**
(DOCX)

**S1 Checklist. PRISMA-P 2015 checklist.**
(PDF)

## Acknowledgments

The authors would like to thank Dr. Ruvistay Gutiérrez for his assistance in developing the search strategy.

## Author Contributions

**Conceptualization:** Felipe Martínez, Catalina Tobar.

**Data curation:** Felipe Martínez.

**Funding acquisition:** Carla Taramasco.

**Investigation:** Carla Taramasco.

**Methodology:** Felipe Martínez, Catalina Tobar.

**Project administration:** Felipe Martínez, Carla Taramasco.

**Software:** Felipe Martínez.

**Supervision:** Felipe Martínez, Carla Taramasco.

**Validation:** Felipe Martínez, Catalina Tobar.

**Writing – original draft:** Felipe Martínez, Catalina Tobar, Carla Taramasco.

**Writing – review & editing:** Felipe Martínez, Catalina Tobar, Carla Taramasco.

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
