## [Decision Letter · Decision Letter 0]

7 Sep 2023

PONE-D-23-16275Effects of Internet-Based Telemonitoring Platforms on the Quality of Life of Oncologic Patients: A Systematic Literature Review ProtocolPLOS ONE

Dear Dr. Martinez,

Thank you for submitting your manuscript to PLOS ONE. After careful consideration, we feel that it has merit but does not fully meet PLOS ONE’s publication criteria as it currently stands. Therefore, we invite you to submit a revised version of the manuscript that addresses the points raised during the review process.

Based on the reviewers` comments and my own consideration of this protocol, this could not be accepted by PLOS ONE in this current format. A major revision is necessary in terms of justification and rational behind conducting this systematic review based on a short synthesis of the available knowledge and a rapid search on the probable reviews indexed and published in Cochrane library, restructuring the protocol considering the discussion is not normally included in the protocol abstract, the necessity of presenting a clear flowchart of the search strategy, or testing the strategy if there is available. The guideline/s used as the reference of conducting the systematic review should be described as well.

We look forward to receiving your revised manuscript.

Kind regards,

Peivand Bastani

Academic Editor

PLOS ONE

Journal Requirements:

2. In your cover letter, please confirm that the research you have described in your manuscript, including participant recruitment, data collection, modification, or processing, has not started and will not start until after your paper has been accepted to the journal (assuming data need to be collected or participants recruited specifically for your study). In order to proceed with your submission, you must provide confirmation.

“no”

Reviewers' comments:

Reviewer's Responses to Questions

**Comments to the Author**

1. Does the manuscript provide a valid rationale for the proposed study, with clearly identified and justified research questions?

Reviewer #1: No

Reviewer #2: Partly

2. Is the protocol technically sound and planned in a manner that will lead to a meaningful outcome and allow testing the stated hypotheses?

Reviewer #1: No

Reviewer #2: Partly

3. Is the methodology feasible and described in sufficient detail to allow the work to be replicable?

Reviewer #1: No

Reviewer #2: Yes

4. Have the authors described where all data underlying the findings will be made available when the study is complete?

Reviewer #1: No

Reviewer #2: No

5. Is the manuscript presented in an intelligible fashion and written in standard English?

Reviewer #1: No

Reviewer #2: Yes

6. Review Comments to the Author

You may also provide optional suggestions and comments to authors that they might find helpful in planning their study.

Reviewer #1: Dear Editorial Office,

I would like to thank you for seeking my opinion on this manuscript.

I went through the whole manuscript and in my reviews I hold this position that the reviewers should help authors to develop their work based on their constructive comments. However, with this submission, the rationale behind conducting the study is not clear, and although a number of objectives set for this study, personally I do not see any particular findings from this study.

Indeed, through reviewing the manuscript, I am not sure if the right approach was addressed in practice, as the organization of the work and the presentation of the material suggest so. For example, no language restriction was applied, now search flow diagram was presented, and no well- structured findings section is presented.

Overall, I regret to have a "NO" with this submission.

Thank you

Sincerely yours

Reviewer #2: Abstract:

- The "Discussion" section is generally not included in the abstract of a systematic review protocol.

Introduction:

- Stating the statistical data from a specific source, such as Chile's Ministry of Health, might not be directly relevant to the broader research synthesis. The inclusion of global statistics will better align with the broader research synthesis being conducted in a systematic review.

- To further strengthen the rationale, consider briefly discussing the limitations or challenges of traditional follow-up methods for cancer patients.

- It might be beneficial to explicitly state the target population (adult cancer patients) to provide a more precise understanding.

- It's important to reference existing reviews in this domain and highlight the contribution of this study and potential research gaps.

Method

- As AMSTAR-2 is primarily a critical appraisal tool for systematic reviews of healthcare interventions, not a guideline for reporting systematic review protocols. It would be more appropriate to emphasize adherence to reporting guidelines like PRISMA-P when describing the protocol's design and methodology.

- The "Eligibility criteria" section should be explicitly outlined as a heading, and it's equally important to include exclusion criteria based on the PICO elements to provide a clear and comprehensive understanding of the study's scope.

- Define all outcomes, including the prioritization of main and additional outcomes, and provide a rationale for this prioritization to ensure transparency and clarity in the study protocol.

- A brief overview of what data will be collected and how it will be organized can help readers understand the data extraction process.

- Quality Assessment in Subgroup: Address whether study quality will impact subgroup analysis.

- Briefly define Review Manager (RevMan) for non-experts.

Discussion:

- A systematic review typically does not include a discussion section.

7. PLOS authors have the option to publish the peer review history of their article (what does this mean?). If published, this will include your full peer review and any attached files.

Reviewer #1: No

Reviewer #2: No

---

## [Author Response · Author response to Decision Letter 0]

30 Sep 2023

Dear Dr. Bastani,

Thank you for your interest in our manuscript and for raising observations that undoubtedly help us improve the quality of our work. Below, we will respond to the commentaries raised by the review team, which we have divided into two sections to facilitate reading.

Reviewer #1

We regret not having generated sufficient interest in reviewer #1.

Regarding the comments briefly mentioned in the reviewer’s objection, we must point out that our proposal corresponds to a research protocol, so it is not yet in a position to report results or findings. Furthermore, it should be considered that the absence of language restrictions in the search process is commonly viewed as a strength when conducting highly sensitive search strategies, as established by the Cochrane Collaboration.

In our opinion, it is appropriate to report the results of a search strategy in a results section of a systematic review, which is not yet applicable to this text as it is a review protocol. However, this phase of the study has already been completed, and we are in a position to supplement the protocol with the results of the literature search. This has been added as a PRISMA flowchart to the study protocol.

Reviewer #2

We appreciate Reviewer #2's time in evaluating our manuscript and pointing out areas for improvement in our work. We’ll address the suggestions and comments that the reviewer provided sequentially:

a) The "Discussion" section is generally not included in the abstract of a systematic review protocol. 

The discussion section has been removed from the abstract.

b) Stating the statistical data from a specific source, such as Chile's Ministry of Health, might not be directly relevant to the broader research synthesis. The inclusion of global statistics will better align with the broader research synthesis being conducted in a systematic review.

We have updated these figures from the introduction to the statistics provided by the World Health Organization regarding cancer on a global level.

c) To further strengthen the rationale, consider briefly discussing the limitations or challenges of traditional follow-up methods for cancer patients. 

A paragraph has been added to the introduction to describe current limitations of traditional follow-up methods.

d) It might be beneficial to explicitly state the target population (adult cancer patients) to provide a more precise understanding. 

We have specified that the systematic review aims to assess the impact of telemonitoring among adult patients with cancer on primary objectives and outcomes. The methodology section specifies that anyone over 18 years old will be considered an adult.

e) As AMSTAR-2 is primarily a critical appraisal tool for systematic reviews of healthcare interventions, not a guideline for reporting systematic review protocols. It would be more appropriate to emphasize adherence to reporting guidelines like PRISMA-P when describing the protocol's design and methodology. 

We appreciate the reviewer's clarification regarding these instruments. We have updated our reporting tool to PRISMA-P and adjusted the reference accordingly. 

f) The "Eligibility criteria" section should be explicitly outlined as a heading, and it's equally important to include exclusion criteria based on the PICO elements to provide a clear and comprehensive understanding of the study's scope. 

We have outlined the eligibility criteria section in the manuscript using a subheading. Additionally, we have provided a more detailed specification of the exclusion criteria for the systematic review and added a summary table for improved clarity in the report.

g) Define all outcomes, including the prioritization of main and additional outcomes, and provide a rationale for this prioritization to ensure transparency and clarity in the study protocol. 

We have prioritized and clearly defined all study outcomes in a new section under the methodology titled 'Study Outcomes.' Additionally, we have provided the reasons behind the prioritization of outcomes in the protocol, as suggested by the reviewer.

h) A brief overview of what data will be collected and how it will be organized can help readers understand the data extraction process. 

A brief introduction has been added regarding the information that will be obtained from each study separately and the purposes these data will serve for the subsequent analysis, to enhance readability of the text.

i) Quality Assessment in Subgroup: Address whether study quality will impact subgroup analysis. 

We specified that a subgroup analysis based on the methodological quality of the included studies will be conducted in the statistical analysis section of the systematic review.

j) Briefly define Review Manager (RevMan) for non-experts. 

A brief description of RevMan has been added to our analysis strategy section. 

k) A systematic review typically does not include a discussion section. 

Our discussion section has been removed from the manuscript. We have replaced it with a “Summary” section to conclude the systematic review protocol.

We sincerely thank the reviewers for their time assessing our study protocol and hope that these modifications address their concerns regarding our proposal. 

Kind regards,

Felipe Martínez on behalf of the authors

---

## [Editor Report · Decision Letter 1]

24 Oct 2023

Effects of Internet-Based Telemonitoring Platforms on the Quality of Life of Oncologic Patients: A Systematic Literature Review Protocol

PONE-D-23-16275R1

Dear Dr. Martinez,

We’re pleased to inform you that your manuscript has been judged scientifically suitable for publication and will be formally accepted for publication once it meets all outstanding technical requirements.

Kind regards,

Peivand Bastani

Academic Editor

PLOS ONE
---

## [Editor Report · Acceptance letter]

31 Oct 2023

PONE-D-23-16275R1 

Effects of Internet-Based Telemonitoring Platforms on the Quality of Life of Oncologic Patients: A Systematic Literature Review Protocol 

Dear Dr. Martínez:

I'm pleased to inform you that your manuscript has been deemed suitable for publication in PLOS ONE. Congratulations! Your manuscript is now with our production department. 

Kind regards, 

on behalf of

Dr Peivand Bastani 

Academic Editor

PLOS ONE